# COVID-19 testing systems and their effectiveness in small, semi-isolated groups for sports events

**Masashi Kamo** [1]*, **Michio Murakami** [2], **Wataru Naito** [1], **Jun-ichi Takeshita** [1], **Tetsuo Yasutaka** [3], **Seiya Imoto** [4]

1 National Institute of Advanced Industrial Science and Technology, Research Institute of Science for Safety and Sustainability, Tsukuba, Ibaraki, Japan, 2 Center for Infectious Disease Education and Research, Osaka University, Suita, Osaka, Japan, 3 National Institute of Advanced Industrial Science and Technology (AIST), Geological Survey of Japan, Research Institute for Geo-Resources and Environment, Tsukuba, Ibaraki, Japan, 4 Division of Health Medical Intelligence, Human Genome Center, The Institute of Medical Science, The University of Tokyo, Minato-Ku, Tokyo, Japan

* masashi-kamo@aist.go.jp

**Data Availability Statement:** A code, which runs with the standard C libraries only, and all data used in this study can be found at https://github.com/

## Abstract

In this study, we quantitatively assessed the effectiveness of systems for COVID-19 testing in small groups of sport teams that are semi-isolated from the general population by countermeasures against infection. Two types of group were assumed, and the dynamics of infection within each group was modeled by using a compartment model of infectious disease. One group (Group A) comprised domestic professional sports teams that play many games over a season while remaining within a relatively small region. Polymerase chain reaction (PCR) tests were routinely conducted once every 2 weeks, and the number of infected individuals that could not be quarantined after identification by testing or checking for symptoms was defined as the risk. The other group (Group B) comprised teams that travel across borders for mass-gathering events like the Olympic and Paralympic Games. The teams were isolated for 2 weeks at their destination; frequent testing and checking for symptoms was conducted, and any infected individuals were quarantined. The number of infected individuals participating in games after the isolation period was defined as the risk. In Group A, the number of infected individuals detected by routinely conducted PCR testing was lower than the number of infected individuals detected by checking for symptoms, indicating that routine testing every 2 weeks was not very effective. In Group B, daily PCR testing was the most effective, followed by daily antigen testing. Dual testing, in which individuals with a positive antigen test were given an additional PCR test, was the least effective with an effect equal to PCR testing every other day. These results indicate that repeated testing does not necessarily increase the detection of infected individuals.

masashi0209/infectious_dynamics_in_small_gourp/.

**Funding:** The author(s) received no specific funding for this work.

**Competing interests:** This study was conducted as part of a comprehensive research project, comprising members from two private companies, Kao Corporation and NVIDIA Corporation, Japan. These members from the companies are not included as the author of this study. W.N. received financial support from the Kao Corporation until March 2020 in a context outside the submitted work. T.Y. and W.N. have received financial support from the Kao Corporation for a collaborative research project in the context of measures at mass-gathering events. T.Y. and W.N. have received financial support from the Yomiuri Giants, the Japan Professional Football League, and the Japan Professional Basketball League. M. K., M.M., T.Y. W.N, and S.I. have attended the New Coronavirus Countermeasures Liaison Council jointly established by the Nippon Professional Baseball Organization and the Japan Professional Football League as experts without any reward. T. Y. and W.N. are advisors to the Japan National Stadium. The findings and conclusions of this article are solely the responsibility of the authors and do not represent the official views of any institution.

## Introduction

Since the outbreak of COVID-19, about 223,022,538 cases have been confirmed worldwide as of 12 September 2021, including about 4,602,882 deaths [1]. Vaccination has led to a decline in the prevalence of the disease, but the rise of variants has led to the spread of the disease again in some countries [2].

Although social measures involving individual behavioral changes such as mask-wearing and lockdowns have been implemented [3], some professions have resumed their socio-economic activities by conducting regular testing for infection, a typical example being professional sports. In domestic sports such as the professional baseball and soccer leagues in Japan and in international sports competitions such as the European Football Championship [4] and the Australian Open Tennis Tournament [5], routine antigen and polymerase chain reaction (PCR) tests and isolation of positive cases and their close contacts are carried out. In the Tokyo Olympic and Paralympic Games that were held July to September 2021, daily antigen testing of athletes and accompanying staff, and additional PCR testing for those identified as positive by the antigen tests, were mandatory [6].

The effectiveness of a test in identifying infected individuals depends on the sensitivity of the test and the frequency of testing [7, 8]. Dickens et al. [9] examined the effectiveness of isolation of infected individuals identified by testing at airports. Panovska-Griffiths et al. [10] investigated the effectiveness of regular testing of the public in reducing the spread of infection and Du et al. [11] estimated the cost-effectiveness of testing and isolation for the public by using well-studied compartment models for infectious disease dynamics (SEIR models [12]) and proposed strategies for effective testing regimens. Ndii and Adi [13] argued efficacy of intervention strategies under a limited resource. Those studies examined the effectiveness of countermeasures in relatively large populations, but no study has examined their effectiveness in small, semi-isolated groups of limited personnel, such as athletes and staff.

It is expected that the optimum testing system would be different between groups that stay in the same region for a long period of time (through a season), as in the case of domestic professional sports leagues, and groups that concentrate in one country for a short period of time and disperse after the games, as in the case of international games such as the Olympic and Paralympic Games. In relation to optimizing disease management in these groups, it is important to quantitatively evaluate how often antigen and PCR testing should be performed. Quantitative evaluation of the effectiveness of the testing system used in the Tokyo Olympic and Paralympic Games will provide important insights for similar mass-gathering events in the future.

The present study evaluated the effectiveness of the current testing systems in two types of small group. In the first group—a small population that plays in a national league several times through a season—we quantitatively evaluated the contribution of PCR testing once every 2 weeks to identifying and isolating infected individuals. In the second group—a small population gathering in one place to play in an international championship—we examined the efficiency of test frequency and test sensitivity (PCR or antigen) in controlling infection, as well as the effect of failed specimen collection. The Tokyo Olympic and Paralympic Games adopted saliva samples for daily antigen testing and additional PCR testing to speed up the process, but these samples are not always accurate and appropriate because they are easily affected by gargling, eating, or drinking immediately before the test; however, the role of such errors on infection control has not been examined at all. In all cases, the population dynamics were analyzed by using a probabilistic compartment model for infectious disease dynamics.

## The model and scenarios for risk evaluation

The model used in this study is a SEIR model containing the numbers of individuals in a population who are susceptible ($S$), exposed ($E$), infected ($I$), pre-symptomatic at day 1 ($P_1$), pre-symptomatic at day 2 ($P_2$), infected and symptomatic ($Is$) or infected and asymptomatic ($Ia$), and recovered ($R$). Two pre-symptomatic states ($P_1$, $P_2$) are considered because test sensitivity for $P$ differs between day 1 and day 2 (it is assumed that an individual spends 2 days in state $P$ on average; see below for parameter settings).

An individual in state $S$ who comes into contact with an infected individual (state $P$, $Ia$, or $Is$) becomes state $E$ at rate β. Individuals in state $E$ are assumed to not be infective. $E$ state individuals become $P_1$ state individuals at rate σ. $P_1$ state individuals become $P_2$ state individuals at rate $\rho_1$. $P_2$ state individuals become either $Ia$ or $Is$ state individuals at rate $\rho_2$. The partition coefficient for states $Ia$ and $Is$ is $\eta$; the probability that a $P$ state individual becomes $Is$ is $\eta$ and the probability that a $P$ state individual becomes $Ia$ is $1 - \eta$. Both $Ia$ and $Is$ state individuals become $R$ state individuals at rate γ.

The dynamics of these states are mathematically described as

$$
\begin{aligned}
\frac{dS}{dt} &= -\beta S(P_1 + P_1 + Ia + Is)/N, \\
\frac{dE}{dt} &= \beta S(P_1 + P_1 + Ia + Is)/N - \sigma E, \\
\frac{dP_1}{dt} &= \sigma E - \rho_1 P_1, \\
\frac{dP_2}{dt} &= \rho_1 P_1 - \rho_2 P_2 \\
\frac{dIa}{dt} &= (1 - \eta)\rho_2 P_2 - \gamma Ia, \\
\frac{dIs}{dt} &= \eta \rho_2 P_2 - \gamma Is
\end{aligned}
\tag{1}
$$

where $N$ is the total number of individuals. Throughout this study, $N = 100$.

These non-linear ordinal differential equations are numerically solved by several ways such as Euler and Lunge-Kutta methods and some other methods [14, 15] in a deterministic manner. In this paper, we solve these equations by stochastic individual based simulation. For example, an $E$ state individual becomes $P$ state with a probability $\Delta t \sigma$ and remains in state $E$ with a probability $1 - \Delta t \sigma$, where $\Delta t$ is a time scaling parameter set at 0.01 throughout the present study, and repetition of the random choice 100 times corresponds to 1-day dynamics.

## Definitions of groups, testing systems, and risk

Players in professional sports teams and in the Olympic and Paralympics Games are under stricter behavior limitations than the general public, and groups of players are semi-isolated from general society. Because perfect isolation is not possible, it can be assumed that infection in a group occurs through rare contact between the players (and staff) and members of the general public. Once infection occurs in a group, it is assumed that the infection spreads within the group according to standard infectious disease dynamics (Eq 1). Because contact between players (and staff) and members of the general public is rare, it is assumed that diseases are brought in from outside the group only once during the period we consider in our simulation (maximum 2 weeks), and the number of infected individuals is only one (initial condition of simulation is $E(0) = 1$, see Eq 1). In reality, there is the chance that two individuals will be infected at the same time or that infections will be brought into the group twice at different

times, and the chance depends on the number of infected individuals in the general public, but in this study we ignore such conditions and we do not consider the dynamics of the general public assuming that the prevalence in the general public is stable in the period we concern for the sake of simplicity.

The aim of the present study is not to investigate the long-term dynamics of COVID-19 but to evaluate the efficiency of testing systems. Japanese professional soccer teams perform routine PCR testing every 2 weeks. Players in groups that travel from abroad are often isolated for 2 weeks. These examples show that 2 weeks is often a unit of measurement for dealing with this disease, and hence in this study we simulate the infection dynamics of the disease for a maximum of 2 weeks.

As stated in the Introduction, we consider two types of group. One is a group like a professional baseball team (Group A). Players in the group are semi-isolated from society. Their movements are limited between home and away fields. The simulation of infection dynamics starts with 1 infected individual (state $E$). During the simulation of up to 2 weeks, a PCR test is done once sometime within the 2 weeks. (The day that the PCR test is conducted cannot be identified. This is because the day the first infected individual appears is a random event, and if we take the day that the first infected individual appears to be the starting point of the simulation, then the day that the PCR test is conducted will be random). Checking for symptoms are conducted daily. On the day of the PCR test, the PCR test is done prior to the checking for symptoms. Infected individuals confirmed by PCR test (individuals in state $P_1$, $P_2$, $Ia$, or $Is$ can be detected) are quarantined and do not contribute to the further spread of the infection both in the team and in the general public until they get recovered. It is assumed that that infected individuals in $Is$ state can always be identified as infected individuals by checking for symptoms and they are quarantined from the group by daily checking. The number of unidentified infected individuals remaining in the group is defined as the risk. In reality, if infected individuals are found, additional testing (PCR or antigen testing) may be done on all players in the group. This leads to a reduction in risk but is not considered in the present study.

In the other group (Group B), the members are almost fully isolated in a "bubble" for 2 weeks after arriving at their destination, as was the case of the Tokyo Olympic and Paralympic Games, and return home after the games. During the 2 weeks of isolation, checking for symptoms and testing is conducted daily. There are three types of test system: an antigen test, a PCR test, and an additional PCR test for individuals who test positive for antigens. Those who are confirmed in any test system to be infected are quarantined from the group. The number of unidentified infected individuals at day 14 (2 weeks) when isolation is ceased is defined as the risk. This is the number of individuals who participate in the games in an infected state, resulting in contact between infected and non-infected individuals across the group. Note that the number in state $E$ is excluded from the number of infected individuals because state $E$ does not cause new infections.

It is assumed that specimens collected for testing can include those with failed collections or subject to unintentional (or intentional) specimen mishandling. The effect that test errors resulting from such specimens has on the risk is considered.

In this group, there is only one event of infection from outside and only one infected individual. The cases to consider are individuals already infected on arrival, and individuals not infected at arrival but newly infected during the isolation period. Let $p_p$ be the probability that there is one infected individual at the point of entry into the destination. Let $p_{pd}$ be the probability that an individual in the group is infected per day after entry into the destination. The probability that one infected individual occurs in a group of $N$ individuals ($p_d$) is given by the

binomial distribution as

$$p_d = \frac{N!}{1!(N-1)!}p_{pd}^1(1-p_{pd})^{N-1} = Np_{pd}(1-p_{pd})^{N-1}. \tag{2}$$

It is assumed that infection is brought in from outside the group just once. The probability that one individual in the group is infected at arrival is $p_p$. The probability that the individual is not infected at arrival $(1-p_p)$ and is newly infected the next day $(p_d)$ is $(1-p_p)\,p_d$, and the probability that the first infection occurs at day 2 is $(1-p_p)(1-p_d)p_d$. The probabilities are generalized to the first infection at day $t$ as

$$p(t) = \begin{cases} p_p & \text{for } t = 0 \\ (1-p_p)(1-p_d)^{t-1}p_d & \text{for } t > 0 \end{cases}. \tag{3}$$

The settings for each group are summarized in Table 1.

## Model parameters

Average duration (in days) at each state ($E$, $P_1$, $P_2$, $Ia$, $Is$) were from He et al. [16] as rounded to the nearest integer, and hence infected individuals stay 3 days as $E$ ($\sigma = 1/3$), 1 day as $P_1$ ($\rho_1 = 1$), 1 day as $P_2$ ($\rho_2 = 1$), 7 days as $Ia$ or $Is$ ($\gamma = 1/7$). The partition coefficient ($\eta$) of $Ia$ and $Is$ states was from He et al. [17] and is 0.54 ($Ia$:$Is$ = 0.46:0.54). The basic reproductive number ($R_0$) of the SARS-CoV-2 Delta variant is found to be 5.08 [18], but the groups that participated in the Olympic and Paralympic Games and professional sports teams are subject to stricter controls than the general public; hence, $R_0$ is assumed to be 4 in this study, although $R_0$ may be even smaller under the condition that most players and staffs have already been vaccinated. Because the average duration in an infective state ($P1$, $P2$, $Ia$, or $Is$) is 9 days, the transmission rate $\beta$ is obtained by solving $4 = 9\beta$, and is $\beta = 4/9$.

Here, the transmission rate was assumed to be constant because of the short period of up to 14 days. In reality, the transmission rate may decrease when an infected person is detected within a close group, which may lead to strong infection prevention measures. Therefore, it is possible that the risk estimated in this study is overestimated. Parameters for sensitivities of the PCR test are $s_1 = 0.33$ (sensitivity with respect to $P_1$), $s_2 = 0.62$ (sensitivity with respect to $P_2$), and $s_3 = 0.80$ (sensitivity with respect to $Ia$ and $Is$) [19]. The sensitivities of the antigen test are 70% of these values of the PCR test [20].

**Table 1. Summary of information about groups.**

| Group definition | |
|---|---|
| Group A | Playing in a domestic league. Travel is limited to domestic travel. |
| Group B | Playing a large international competition in which several teams gather. Travel is across the border. |
| Testing system | |
| Group A | PCR test every 2 weeks. Daily checking for symptomatic individuals. |
| Group B | Daily antigen test or PCR test or both. Daily checking for symptomatic individuals. |
| Simulation length | |
| Group A | Maximum 2 weeks. If positive confirmation is given, simulation is complete. |
| Group B | Two weeks |
| Risk definition | |
| Group A | Number of infected individuals remaining in the population after removal of those who tested positive. |
| Group B | Number of infected individuals remaining at the end of the 2-week isolation period (number of infected individuals participating in the games) |

In the test system whereby individuals who test positive for antigens have an additional PCR test, two tests are conducted within a very short time. The total sensitivities are assumed to be multiples of the sensitivities of the antigen and PCR tests (i.e., assumption of independent sensitivities). In reality, because the test sensitivities are dependent on viral load, it is likely that individuals who test positive for antigens will be positive for the second PCR test, hence the test's power may be estimated to be lower than it actually is under the assumption of independent sensitivities.

The probability that a group in Group B has one individual who is infected at arrival ($p_p$) is $1.0 \times 10^{-4}$. The probability depends on the infection prevalence and management system in the country of origin, and hence the value is based on an assumption. The probability that an individual in a group is infected per day ($p_{pd}$) is $1.0 \times 10^{-6}$, and this is also based on an assumption. Under this value for $p_{pd}$, the probability that one individual in a group of 100 players becomes infected per day ($p_d$) is about $1.0 \times 10^{-4}$ (see Eq 2). Under these assumptions, the values of $p_p$ and $p_d$ are almost identical, implying that even before departure the group is under the same level of control as that at the destination (at the destination, the group is isolated in a bubble, and hence it is likely that $p_d < p_p$). These parameters are summarized in Table 2.

In both testing scenarios, we performed 10,000 simulations and examined the average of the total number of infected individuals.

## Results

### Group A: The effect of PCR testing every 2 weeks

Fig 1 shows the infection dynamics for a group of 100 players without any countermeasures against infection (no checking for symptoms, no tests at all). The total number of infected individuals at day 9 is about 4 (agreeing well with the assumption of $R_0 = 4$), and the total number of infected individuals at about day 14 is about 10, implying that 10% of the individuals in a group are infected in 2 weeks, if no countermeasures are implemented.

If the starting point of the simulation is taken to be the timing that an individual in the group becomes infected, then the day on which PCR testing is conducted is a random event, and the tests are conducted on day 0 to 13 with equal probability. Simulations were performed for each of all possible days with 10,000 replicates. The results are summarized in Table 3.

Fig 2A shows whether the detection of infected individuals was made by routine testing or by checking for symptomatic, for each day the test was conducted. When the first infected

Table 2. Summary of model parameters.

| Parameters | Values | Notes |
|---|---|---|
| $\sigma$ | 1/3 | Transition rate from state $E$ to state $P_1$ per day |
| $\rho_1$ | 1 | Transition rate from state $P_1$ to state $P_2$ per day |
| $\rho_2$ | 1 | Transition rate from state $P_2$ to state $Ia$ or $Is$ |
| $\eta$ | 0.54 | Fraction that $P_2$ becomes state $Is$ |
| $\gamma$ | 1/7 | Transition rate from $Ia$ or $Is$ to $R$ (recovery rate) |
| $\beta$ | 4/9 | Transmission rate per day per contact |
| $s_1$ | 0.33 | PCR test sensitivity to state $P_1$ |
| $s_2$ | 0.62 | PCR test sensitivity to state $P_2$ |
| $s_3$ | 0.80 | PCR test sensitivity to state $Ia$ and $Is$ |
| $p_p$ | $1.0 \times 10^{-4}$ | Probability that an individual is infected at arrival |
| $p_{pd}$ | $1.0 \times 10^{-6}$ | Probability that an individual is infected per day |
| $N$ | 100 | Number of individuals in a group |

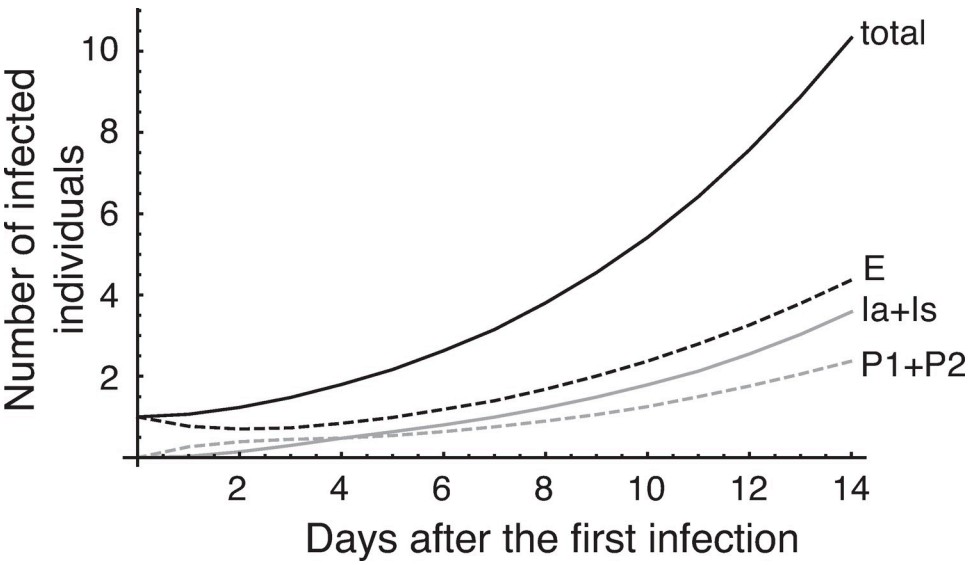

**Fig 1. Infection dynamics without any countermeasures.** The initial values are $S(0) = 99$, $E(0) = 1$, and the others are 0. The average of 10,000 replicates. Total number of infected individuals is the sum of $E$, $P_1$, $P_2$, $Ia$, and $Is$.

individual appeared on the day that PCR testing was conducted (day 0 on the horizontal axis), all detection of infected individuals was made by checking for symptoms (a black dot). The detection rate by PCR test (gray dots) was always lower than that by the checking for symptoms. When the PCR testing was conducted at day 4 after the first infection, about 45% of infected individuals were detected by the test (about 55% were detected by the checking for symptoms). This result implies that the infected individuals were mostly detected through the checking for symptoms rather than by the PCR test.

Fig 2B shows the average waiting time until infected individuals were detected. If an infected individual appeared in the group on the day of the PCR test (day 0), it took an average of 6.5 days to detect subsequently infected individuals, and all detection was made by checking for symptomatic individuals. If the PCR test was conducted 4 days after the first infected individual appeared in the group, the average waiting time was shorter than that of testing at day 0. This is because the detection by PCR test was maximum (Fig 2A), but the difference from testing at day 0 is just about 1 day.

The fraction of infected individuals that were detected by checking for symptoms was 73.13% [= 100 × 62.50 / (62.50 + 22.97)] (Table 3). In this test system, it can be concluded that

**Table 3. Summary of simulation results.** Averages of 10,000 repetitions each from day 0 to day 13 when the test is performed (140,000 replicates total).

| | |
|---|---|
| Average days until the first positive confirmation (day) [1] | 6.117 |
| Number of positive individuals quarantined from the population due to positive confirmation [1] | 1.147 |
| Number of infected individuals remaining in the group after quarantine of detected infected individuals (number among them with $E$ status).[1] This is the risk. | 1.970 (1.211) |
| Percentage of 140,000 repetitions in which positive cases were confirmed by symptom checking (%) [2] | 62.50 |
| Percentage of 140,000 repetitions in which positive cases were confirmed by PCR testing (%) [2] | 22.97 |
| Percentage of 14-day simulations with no positive confirmation (%) [2] | 14.53 |

[1] Averages exclude cases where infected individuals were not confirmed within 14 days of simulation

[2] Sum of these three percentages becomes 100%

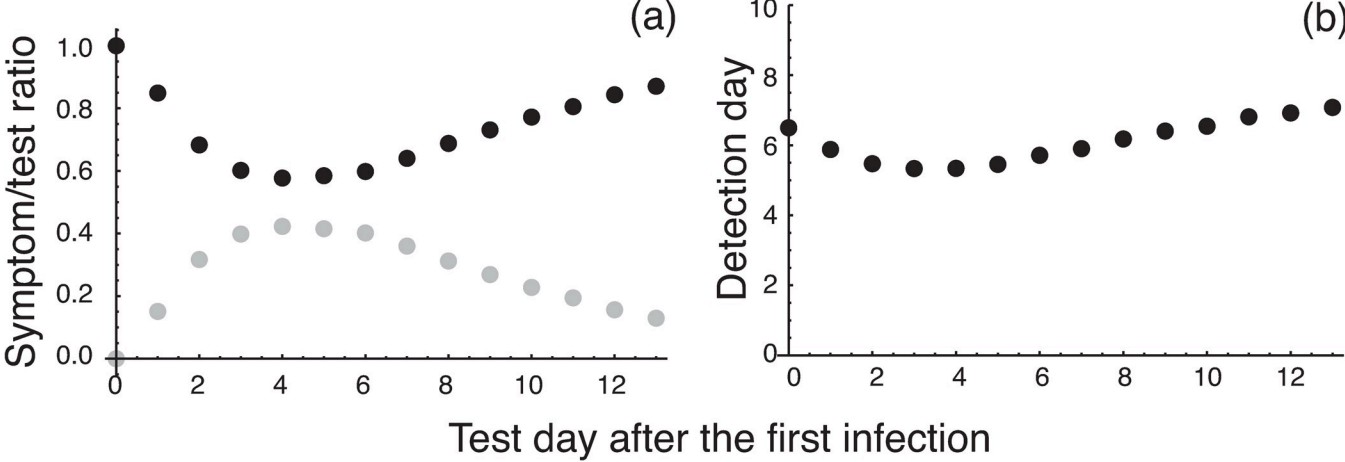

**Fig 2.** Detection rate of infected individuals (a) and waiting time for infection detection (b). (a) The ratio of infected individuals detected by checking for symptoms (black dots) with respect to those detected by routine PCR testing (gray dots). (b) The average waiting time until detection of infected individuals after the first infection occurred in the group. The overall average is 6.1 days. This implies that the first infected individual in the group occurs about 6 days (on average) before subsequently infected individuals are found.

the role that routine PCR testing plays in detecting infected individuals is not as significant as checking for symptoms.

### Group B: The effect of daily tests and daily checking for symptomatic individuals

Fig 3 shows the infection dynamics for a test system involving a daily antigen test and additional PCR test for those who test positive for antigens (the number of infected individuals

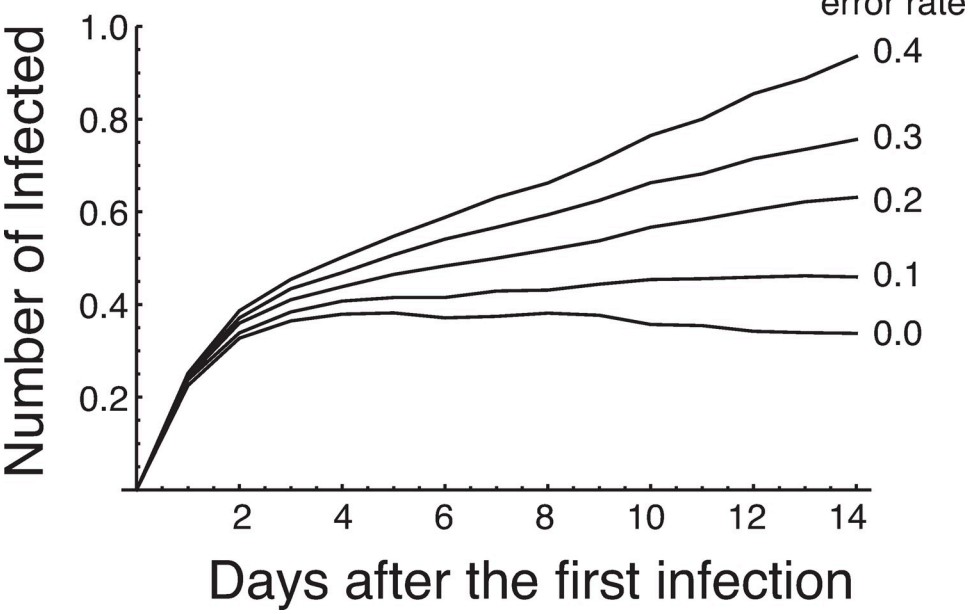

**Fig 3. Infection dynamics in Group B with the test system including an additional PCR test for individuals who test positive for antigens.** The initial condition of the dynamics is $S(0) = 99$, $E(0) = 1$, and all others are 0. The number of infected individuals in state $E$ is excluded in the figure (see Eq 4 for the number of infected individuals). This is because individuals in state $E$ are not infective and have no effect even if they participate in games.

excluding the individuals in state $E$ are shown). Infected individuals were quarantined from the group, and the simulation continued for 14 days. The dynamics with various test error rates are shown. When there is no error, the number of infected individuals rises once but then decreases at around day 3 or 4. The dynamics indicates that the effective reproductive number is less than 1 in this test system with no error. In contrast, when the error rates are above 10%, the number of infected individuals keeps rising, indicating that the effective reproductive number is greater than 1.

We assume that an infectious disease is brought into the group from the outside only once. Under this assumption, if the infection is brought into the group 2 days after arrival at the destination, the infection spreads within the group for 12 days. The number of infected individuals is given by the number at day 12 in Fig 3. We define the total number of individuals infective at day $t$ after the first infection as

$$I(t) = P_1(t) + P_2(t) + Ia(t) + Is(t), \qquad (0 \leq t \leq 14). \qquad (4)$$

The probability that infection is brought into the group at day $t$ is given by Eq 3, and hence the average number of infected individuals playing in the games ($I_m$) is

$$I_m = \sum_{t=0}^{14} p(t) I(14 - t). \qquad (5)$$

The dynamics of the test system with antigen and PCR tests can be solved in the same way as in Fig 3. Using the result of the dynamics and Eq 5, the risk of infected individuals participating in games after isolation can be obtained.

The results are shown in Fig 4A. Although it is almost obvious, the risk becomes higher as the error rate rises in any system. The risk is the lowest in the test system with PCR testing only, the second lowest in the test system with antigen testing only, and highest in the test system with an additional PCR test for individuals testing positive for antigens. The combination of two tests leads to increased risk, because infection is determined only when both tests are positive. This order was kept for all test error rates except when the rate was 1.0 which

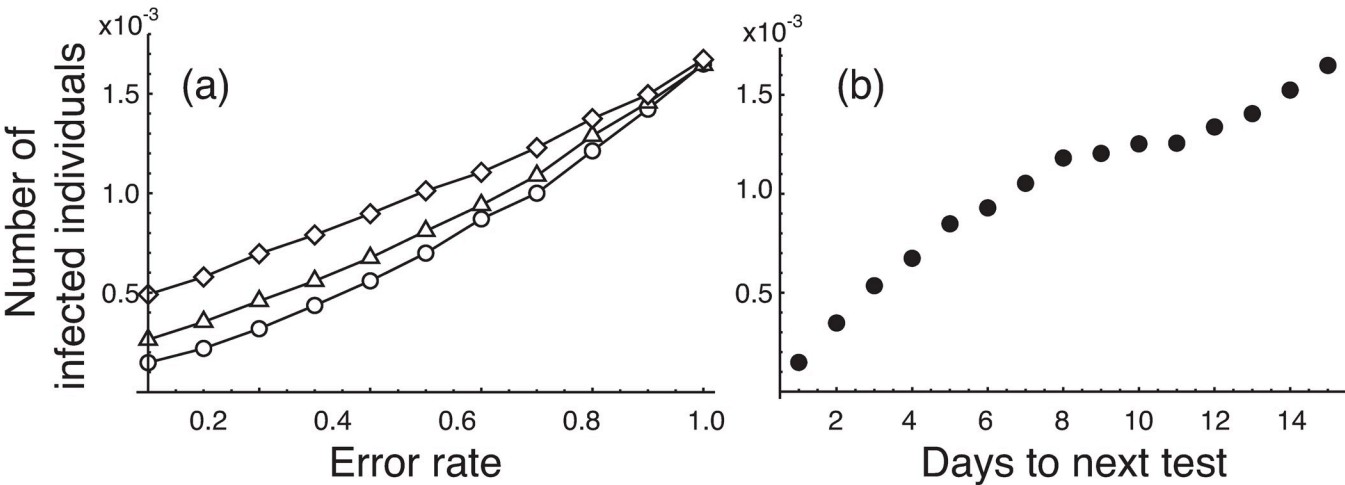

**Fig 4. Risks in Group B in various measurement scenarios.** (a) Risks in Group B with different test systems and different test error rates. The risk with the test system with daily PCR testing only (○) was the lowest, that with daily antigen testing only (△) was the second lowest, and that with an additional PCR test for individuals testing positive for antigens (◇) was the highest. In all simulations, infected individuals showing symptoms (state $Is$) are quarantined daily. (b) The risk with different test frequency. Only PCR testing with 0% error was used for the test. Numbers on the horizontal axis indicate the number of days to the next test. For example, a number 2 means that the test frequency was every 2 days. In all test frequencies, the first test on arrival (day 0) at the destination is conducted at day 0. At up to 7-day frequencies, more than two tests were conducted within the 14 days' isolation. After 8 days, tests were conducted two times, and at 15 days, testing is only once at day 0.

corresponds to no tests (ideally, risk by all test systems should be the same at this situation, but they were different due to variability caused by stochastic simulations).

When the error rate is 0, the risk of the test system with an additional PCR test for individuals testing positive for antigens was about $0.5 \times 10^{-3}$. In the test system with the PCR test only, the risk reaches $0.5 \times 10^{-3}$ when the error rate is about 0.4. Thus, the risk with the test system with an additional PCR test for individuals testing positive for antigens with zero error rate is equivalent to the risk with the test system that with a PCR test only with error rate of 0.4.

Fig 4B shows the risk by only PCR testing at different test frequencies, with test error rate of zero. Not surprisingly, the risk increased as the test frequency was reduced. Numbers along the horizontal axis represent the days to the next test, and hence a number 1, for example, means that the test is conducted daily. The risk for the "1 day to next test" in Fig 4B is equal to the risk with zero error rate of the daily PCR test in Fig 4A (∘ on the vertical axis). The risk for the "15 days to next test" in Fig 4B is equal to the risk with 100% error rate, implying no test in Fig 4A. In all results in Fig 4B, PCR tests were conducted at least once on arrival (day 0) at the destination. The reason why the number of tests is different but the risk is the same is that the individuals already infected on arrival at the destination are in state $E$, which is a state that cannot be detected by testing. Furthermore, by comparing Fig 4A and 4B, we can understand the risk equivalency between test error and test frequency. In the test system with PCR only (∘ in Fig 4A), the risk was $0.319 \times 10^{-3}$ when the error rate was 0.2. The risk with PCR testing every 2 days was $0.347 \times 10^{-3}$. These values imply that daily PCR testing with 20% error and PCR testing every 2 days with 0% error are risk equivalent.

The number of infected individuals at each time $t$ in Eq 5 is shown in a stacked bar graph in Fig 5. This figure shows the contribution of the day the infected individual appeared in the group to the risk. In addition to the three test systems shown in Fig 4A, the risk with no tests but with daily checking for symptoms is also shown. Fig 5A shows risks with $p_P$ (probability that an individual is infected on arrival) is $1.0 \times 10^{-4}$ (the value we have investigated so far, see Table 2). Fig 5B shows risks with $p_P = 1.0 \times 10^{-3}$ (10 times higher). In Fig 5A, the risk without testing was $1.673 \times 10^{-3}$. By conducting tests, the risk became $0.491 \times 10^{-3}$ (antigen+PCR), $0.263 \times 10^{-3}$ (antigen only), and $0.1480 \times 10^{-3}$ (PCR only), and rates of reduction were respectively 70.65%, 84.25%, and 91.15%. When $p_P$ is 10 times higher (Fig 5B), risk without testing was $3.78 \times 10^{-3}$, and the risk was reduced by $0.795 \times 10^{-3}$ (antigen+PCR), $0.374 \times 10^{-3}$ (antigen only), and $0.189 \times 10^{-3}$ (PCR only). The contribution of $p_P$ (the brightest gray area) on all risks with no tests in Fig 5A was just 13.00%, but the contribution was 61.95% in Fig 5B. These results indicate that it is important to keep the risk of infection as low as possible at the point of arrival at the destination.

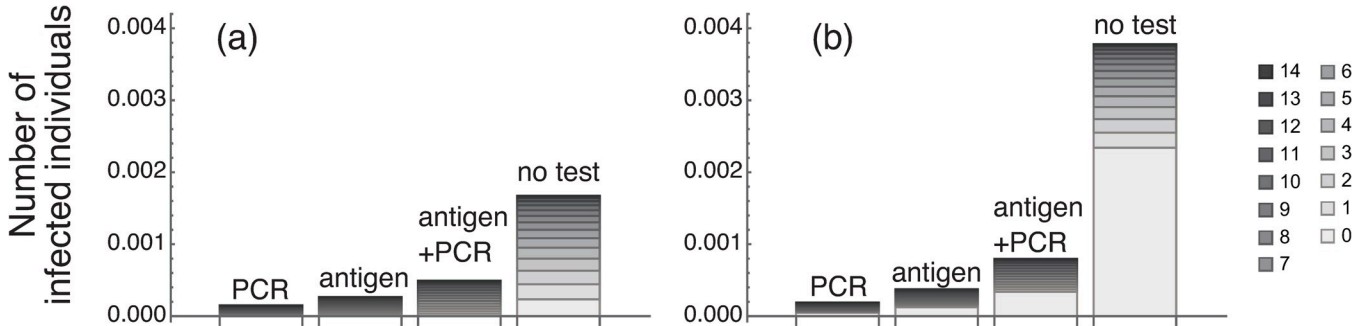

**Fig 5. The contribution of the day on which the infected individual appeared.** (a) Results with parameters in Table 2. (b) Results when the probability that an individual was infected at arriving ($p_P$) was 10 times higher (other parameters were the same). Test error rates were assumed to be 0%.

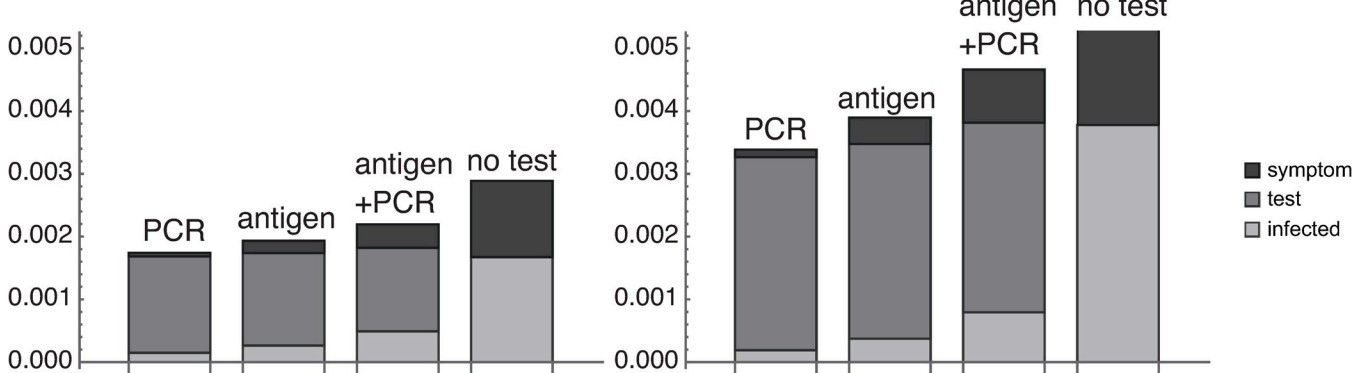

**Fig 6. Number of infected and quarantined individuals.** Number of infected individuals playing in the games (in Fig 5) plus the number of individuals quarantined from the group owing to infection. Black shows the number quarantined by checking for symptom, dark gray shows the number quarantined by routine testing, and light gray is the number in Fig 5 (number of infected individuals participating in the games). In the left panel, the probability that there is an infected individual at arrival ($p_p$) is $1.0 \times 10^{-4}$, and in the right panel the probability is $1.0 \times 10^{-3}$.

So far, we have defined the risk as the number of infected individuals participating in games after 14 days' isolation. The number of individuals quarantined from the group as infected during the 14 days' isolation (the number of individuals who cannot participate in games) might also be considered as a risk. Fig 6 shows the results of adding the number of individuals quarantined from the group due to the infection during the 14 days' isolation to the number of infected individuals participating in the games in Fig 5. In contrast to Group A, removal due to infection mainly occurs by testing positive (except in the case of no tests, where of course there is no positive test result). The numbers of infected persons during the 14 days (i.e., the number of infected individuals playing in the games plus the number of individuals quarantined from the group) were $1.74 \times 10^{-3}$, $1.93 \times 10^{-3}$, $2.20 \times 10^{-3}$, and $2.89 \times 10^{-3}$ for PCR only, antigen only, antigen + PCR, and no test, respectively with $p_p = 10^{-4}$. If the probability that there is an infected individual on arrival increases tenfold, the numbers of individuals quarantined because of infection become about double in all test systems and were $3.38 \times 10^{-3}$, $3.89 \times 10^{-3}$, $4.66 \times 10^{-3}$, and $6.94 \times 10^{-3}$ for PCR only, antigen only, and antigen + PCR, and no test, respectively.

## Discussion

The risk of infectious diseases and the effectiveness of countermeasures in two sports groups of different nature were investigated by using a stochastic compartment model. One group (Group A) is a professional sports team, which spends a season playing several games within a relatively small area. The other (Group B) is a group of players leaving their country to play an international match in a certain destination. They cross a border and are isolated there for a while, have a limited number of games, and then go home just after the games are over.

### Risk in Group A

The countermeasures for the group were a regular PCR test (every 2 weeks) for all players (and staff) and checking for symptomatic individuals. The individuals identified as infected (either by test or by checking for symptoms) are quarantined from the group. After the removal of these identified individuals, some unidentified infected individuals remain in the group, and the number of remaining infected individuals was defined as the risk in this group.

In this group, most of the infected individuals were identified by checking for symptoms, and the efficiency of the PCR test once per 2 weeks in identifying infected individuals was only about 1/3 of that of checking for symptoms (Table 3), indicating that the efficiency of the PCR tests to detect the infected individuals was not high. The reason is almost obvious: testing every 2 weeks is not frequent enough. In our parameterization, a susceptible individual becomes uninfected again 12 days after infection, and hence the dynamics of this infection has 12-day cycles. Identification of infected individuals by routine PCR testing can be thought of as a kind of sample survey. In order to reproduce the 12-day cyclic dynamics, samples must be taken at least every 6 days [21]. In other words, for higher efficiency, the frequency of testing should be increased.

In the test system in Group A, as shown in Table 3, after removal of infected individuals detected by testing and symptom confirmation, an average of 1.970 infected individuals remains in the group, of which 1.211are in state $E$, which cannot be detected by tests. Additional testing may be useful to detect these remaining infected individuals. However, because most of the remaining infected individuals in the group are of $E$ status, few infected individuals would be detected by immediate additional testing. It may be more effective to wait a few days (for state $E$ to become state $P$) before conducting additional tests after strict isolation for preventing new infections while waiting.

## Risk in Group B

In this population, we defined the risk as the number of infected individuals remaining in the group at the end of 2 weeks of isolation (the number of individuals participating in games in an infected state) and calculated the risk under several test systems. As can be seen from Fig 3, when there are no test errors, the effective reproduction number is below 1, and if a test system of daily antigen testing plus additional PCR testing for individuals who test positive for antigens (dual test system) is used, an outbreak of infection within the group can be prevented. However, if the error rate is more than 10%, the effective reproduction number will be more than 1, and the infection will spread within the group.

Among the three test systems (PCR test only, antigen test only, and dual test), the dual test system had the highest risk. Although PCR testing is costly and the number of tests is often limited, antigen testing is prone to false positives. The dual testing system helps reduce the number of PCR tests while also reducing the number of false positives that can occur with antigen testing. At the same time, however, a false-negative result on the second PCR test leads to the missing of infected individuals detected by the first antigen test. The second PCR test can serve as a relief measure for individuals who are identified as infected by false-positive results on the first antigen test. Providing relief measures for false-positive individual is an important task, but because relief measures increase the risk of infection, it is not easy to find out what the best system is. Further careful discussion on this point is needed. Note that we have assumed that the sensitivity of dual testing is a simple multiplication of the sensitivities of these two tests (assuming that the sensitivities of the tests are independent), but this assumption may not be valid in some cases (for example, where there is a possibility that a positive case by the first antigen test is likely to be positive again by the second PCR test).

In Fig 4B, we examined how the risk changed when the frequency of testing was reduced. The risk (the number of infected individuals playing games) was $0.347{\times}10^{-3}$ when the test frequency was every 2 days. The risk with dual testing with no error (◇ in Fig 4A) was $0.491{\times}10^{-3}$, and the value is higher than the risk with PCR testing every 2 days ($0.347{\times}10^{-3}$). This implies that PCR testing every 2 days had a lower risk than daily dual testing. If PCR testing can be performed on all subjects in terms of cost and availability of testing resources, then

performing only the PCR test thoroughly on the subjects may be the most promising option for reducing the risk of infection.

Comparison of Fig 4A and 4B indicates that the risk with daily PCR testing only at an error rate of 20% was similar to the risk with PCR testing every 2 days at an error rate of 0%. This result shows the importance of collecting accurate and appropriate samples. Finding such a risk equivalence relationship is important for decision making, not only for infectious disease control but also for general risk assessment studies.

Daily testing reduced the number of infected individuals after 14 days (i.e., those who participate in games while infected) by nearly 80% (Fig 5, dual test with $p_p = 1.0 \times 10^{-3}$) and reduced the number of infected individuals during 14 days by nearly 30% (Fig 6, dual test with $p_p = 1.0 \times 10^{-3}$), highlighting that daily testing is effective in reducing the number of infected players who go to games. Furthermore, keeping the initial number of infected individuals low is important to reduce the risk. The result is consistent with the result by Ndii et al. [22] who found that finding infected individuals as early as possible is important to reduce the number of undetected infected individuals. If infection control measures are taken before departure, any infected individuals will arrive at the destination in state $E$, a state that cannot be detected by tests. Although it is very important to reduce the initial number of infected individuals, testing immediately after entry into the country is not very useful. At the Tokyo Olympic and Paralympic Games, the athletes were generally tested twice, 96 hours before departure and once on the arrival. They were allowed to participate in Games-related activities for the first 3 days after their arrival if they tested negative for COVID-19 every day and operated under a higher level of supervision. Thorough testing before and after the arrival may have been a useful approach to reduce the risk of infection in the athlete village, but it will need to be verified separately whether only the 3-day testing and supervision after arrival was sufficient.

## Author Contributions

**Conceptualization:** Michio Murakami, Tetsuo Yasutaka, Seiya Imoto.

**Investigation:** Masashi Kamo, Michio Murakami, Jun-ichi Takeshita.

**Project administration:** Seiya Imoto.

**Supervision:** Wataru Naito.

**Writing – original draft:** Masashi Kamo.

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
