## [Decision Letter · Decision Letter 0]

26 Dec 2021

PONE-D-21-32760COVID-19 testing systems and their effectiveness in small, semi-isolated groups for sports eventsPLOS ONE

Dear Dr. Kamo,

Thank you for submitting your manuscript to PLOS ONE. After careful consideration, we feel that it has merit but does not fully meet PLOS ONE’s publication criteria as it currently stands. Therefore, we invite you to submit a revised version of the manuscript that addresses the points raised during the review process.

We look forward to receiving your revised manuscript.

Kind regards,

Asep K. Supriatna, Ph.D

Academic Editor

PLOS ONE

3. Thank you for stating the following in the Competing Interests/Financial Disclosure* (delete as necessary) section:

“This research project includes other members from two private companies, Kao Corporation and NVIDIA Corporation, Japan. W.N. received financial support from the Kao Corporation until March 2020 in a context outside the submitted work. T.Y. and W.N. have received financial support from the Kao Corporation for a collaborative research project in the context of measures at mass-gathering events. T.Y. and W.N. have received financial support from the Yomiuri Giants, the Japan Professional Football League, and the Japan Professional Basketball League. M.K., M.M., T.Y. W.N, and S.I. have attended the New Coronavirus Countermeasures Liaison Council jointly established by the Nippon Professional Baseball Organization and the Japan Professional Football League as experts without any reward. T.Y. and W.N. are advisors to the Japan National Stadium. The findings and conclusions of this article are solely the responsibility of the authors and do not represent the official views of any institution.”

We note that one or more of the authors are employed by a commercial company: Kao Corporation and NVIDIA Corporation

Additional Editor Comments :

Dear Authors,

Please find the attached review reports.

Both the reviewers found that the manuscript is technically sound and well-written. However, they suggested for some minor corrections. Please do the correction by addressing the comments of the reviewers.

Reviewers' comments:

Reviewer's Responses to Questions

**Comments to the Author**

1. Is the manuscript technically sound, and do the data support the conclusions?

Reviewer #1: Yes

Reviewer #2: Yes

2. Has the statistical analysis been performed appropriately and rigorously? 

Reviewer #1: I Don't Know

Reviewer #2: Yes

3. Have the authors made all data underlying the findings in their manuscript fully available?

Reviewer #1: No

Reviewer #2: Yes

4. Is the manuscript presented in an intelligible fashion and written in standard English?

Reviewer #1: Yes

Reviewer #2: Yes

5. Review Comments to the Author

Reviewer #1: COVID-19 testing systems and their effectiveness in small, semi-isolated groups for sports events

This paper investigates the dynamics of COVID-19 within two groups of professional players. There are several comments that needs to be addressed before it has been published.

1. Every players has been tested every two weeks, where is the parameter in the model representing this fact?

2. Line 133-135, when the detected individuals have been confirmed they have been removed from the group. Why are that? Where does it represent in the model? Does it affect the population numbers?

3. They are two groups. Is there any interaction between these two groups and do the authors analyse differently for transmission dynamics of each group?

4. The authors can discuss the following papers

• An analysis of covid-19 transmission in indonesia and saudi arabia. Communication in Biomathematical Sciences. 2020

• Modelling the transmission dynamics of COVID-19 under limited resources. Communication in Mathematical Biology and Neurosciences. 2020.

Reviewer #2: In this paper, the authors have quantitatively assessed the effectiveness of systems for COVID-19 testing in small groups of sport teams that are semi-isolated from the general population by countermeasures against infection. They have assumed two types of group, and the dynamics of infection within each group has modeled by using a compartment model of infectious disease. I think that the paper entitled "COVID-19 testing systems and their effectiveness in small, semi-isolated groups for sports events" is well-organized and the significance of the main ideas are attractive. However, it needs some revisions from the point of authors and readers to improve the quality of the paper. After these major revisions, I suggest that this paper can be accepted to publish in "PLOS ONE".

My other comments are as follows:

1. The authors have used “Figure” in Pages 10, 12 etc. to write some details. Since the word “Figure” are usually use to show plots, it is better to use other title for this end.

2. Also, I consider the literature is not enough and that is why, I recommend the authors to refer to other recent works indexed in Web of Science, Scopus, Emerald, Cambrige, and of course MDPI Journals. For instance:

• Shokri, A., Mehdizadeh Khalsaraei, M., Molayi, M., Nonstandard Dynamically Consistent Numerical Methods for MSEIR Model, J. Appl. Comput. Mech., 8(1), (2022): 196-205.

• Mehdizadeh Khalsaraei, M., Shokri, A., Noeiaghdam, S., Molayi, M., Nonstandard Finite Difference Schemes for an SIR Epidemic Model, Mathematics, 9(23), (2021): 3082.

With many thanks and best regards.

6. PLOS authors have the option to publish the peer review history of their article (what does this mean?). If published, this will include your full peer review and any attached files.

Reviewer #1: No

Reviewer #2: No

---

## [Author Response · Author response to Decision Letter 0]

20 Feb 2022

First of all, we would like to thank both reviewers for reading our paper. We are happy to find that both reviewers recommend our paper for publication after revision. We have revised our paper based on the comments by reviewers. Our response to the comments is as follows.

Due to the request by PLOS ONE, a source code and all data analyzed in the study were presented in an open repository. In the previous version, random numbers were generated by run1() from the Numerical Recipes in C (Press, Teukolsky, Vetterling and Flannery 1992, Cambridge University Press) for Monte-Carlo simulations. The code is a copyrighted material, and hence we changed the random generator to that of the standard C library in the source code in the repository. We’ve redone all the simulations with the new code, and found that the results were consistent with the previous ones (except Figure 1). However, because this is a stochastic simulation, all values are slightly different, and hence we updated all mean values in the text and figures in the revised version. Please find the comparison of figures at the end of this document. 

Reviewer #1: COVID-19 testing systems and their effectiveness in small, semi-isolated groups for sports events

This paper investigates the dynamics of COVID-19 within two groups of professional players. There are several comments that needs to be addressed before it has been published.

We appreciate your constructive comments. 

The comments in 1-3 seem to be questions arising from our lack of description. We solve the dynamics in a group (termed “team”) and we do not consider the dynamics in the general public. To make the point clear, we revised a sentence in page 6, line 124-128

OLD: 

In reality, there is the chance that two individuals will be infected at the same time or that infections will be brought into the group twice at different times, but in this study we ignore such conditions for the sake of simplicity.

NEW:

In reality, there is the chance that two individuals will be infected at the same time or that infections will be brought into the group twice at different times, and the chance depends on the number of infected individuals in the general public, but in this study we ignore such conditions and we do not consider the dynamics of the general public assuming that the prevalence in the general public is stable in the period we concern for the sake of simplicity. (added sentence are shown in italic)

1. Every players has been tested every two weeks, where is the parameter in the model representing this fact?

There are no explicit parameters for testing. The effect of testing on the infectious disease dynamics depends on [1] test sensitivities and parameters for the sensitivities are shown in the text (line 202-204 ) and Table 2, and [2] reduction of infected individuals detected by tests. In the dynamics, the reduction is implemented by decreasing infected individuals minus detected individuals by testing. 

2. Line 133-135, when the detected individuals have been confirmed they have been removed from the group. Why are that? Where does it represent in the model? Does it affect the population numbers?

I regret that the term "remove" were misleading. In the simulation, the detected infected individuals are quarantined while they are infected (the total number of individuals in the team are decreased while they are quarantined). We added a sentence to make the point clear as 

ADDED SENTENCE: 

quarantined and do not contribute to the further spread of the infection both in the team and in the general public until they get recovered (line 144-146).

All the term “removed” in the text were changed to “quarantined”. 

3. They are two groups. Is there any interaction between these two groups and do the authors analyse differently for transmission dynamics of each group?

Since we consider the infectious dynamics in small group only, we do not consider an interaction among groups. We think that the revisions we've made so far solve the concern by the reviewer. 

4. The authors can discuss the following papers

• An analysis of covid-19 transmission in indonesia and saudi arabia. Communication in Biomathematical Sciences. 2020

• Modelling the transmission dynamics of COVID-19 under limited resources. Communication in Mathematical Biology and Neurosciences. 2020.

Thank you very much for these information. We cited the second paper in Introduction (line 57-58) and the first paper in Discussion (line 466-468). 

Reviewer #2: In this paper, the authors have quantitatively assessed the effectiveness of systems for COVID-19 testing in small groups of sport teams that are semi-isolated from the general population by countermeasures against infection. They have assumed two types of group, and the dynamics of infection within each group has modeled by using a compartment model of infectious disease. I think that the paper entitled "COVID-19 testing systems and their effectiveness in small, semi-isolated groups for sports events" is well-organized and the significance of the main ideas are attractive. However, it needs some revisions from the point of authors and readers to improve the quality of the paper. After these major revisions, I suggest that this paper can be accepted to publish in "PLOS ONE".

Thank you very much for your time to read our paper. 

My other comments are as follows:

1. The authors have used “Figure” in Pages 10, 12 etc. to write some details. Since the word “Figure” are usually use to show plots, it is better to use other title for this end.

We agree that the point makes sense, but since “Figure” is specified to use for any graphics and paintings in the PLOS ONE guidance, and we would like to use the term. 

2. Also, I consider the literature is not enough and that is why, I recommend the authors to refer to other recent works indexed in Web of Science, Scopus, Emerald, Cambrige, and of course MDPI Journals. For instance:

• Shokri, A., Mehdizadeh Khalsaraei, M., Molayi, M., Nonstandard Dynamically Consistent Numerical Methods for MSEIR Model, J. Appl. Comput. Mech., 8(1), (2022): 196-205.

• Mehdizadeh Khalsaraei, M., Shokri, A., Noeiaghdam, S., Molayi, M., Nonstandard Finite Difference Schemes for an SIR Epidemic Model, Mathematics, 9(23), (2021): 3082.

Thank you very much for these information. We have cited these papers (line 107-109) and other papers in the revised version. 

With many thanks and best regards.

We too have many thanks and best regard to you.

---

## [Decision Letter · Decision Letter 1]

16 Mar 2022

COVID-19 testing systems and their effectiveness in small, semi-isolated groups for sports events

PONE-D-21-32760R1

Dear Dr. Kamo,

We’re pleased to inform you that your manuscript has been judged scientifically suitable for publication and will be formally accepted for publication once it meets all outstanding technical requirements.

Kind regards,

Asep K. Supriatna, Ph.D

Academic Editor

PLOS ONE

Reviewers' comments:

Reviewer's Responses to Questions

**Comments to the Author**

1. If the authors have adequately addressed your comments raised in a previous round of review and you feel that this manuscript is now acceptable for publication, you may indicate that here to bypass the “Comments to the Author” section, enter your conflict of interest statement in the “Confidential to Editor” section, and submit your "Accept" recommendation.

Reviewer #1: All comments have been addressed

Reviewer #2: All comments have been addressed

2. Is the manuscript technically sound, and do the data support the conclusions?

Reviewer #1: Yes

Reviewer #2: Yes

3. Has the statistical analysis been performed appropriately and rigorously? 

Reviewer #1: N/A

Reviewer #2: Yes

4. Have the authors made all data underlying the findings in their manuscript fully available?

Reviewer #1: No

Reviewer #2: Yes

5. Is the manuscript presented in an intelligible fashion and written in standard English?

Reviewer #1: Yes

Reviewer #2: Yes

6. Review Comments to the Author

Reviewer #1: The authors have appropriately addressed all my concern and the paper can be published in its current form.

Reviewer #2: The reviewer believes that the paper entitled “COVID-19 testing systems and their effectiveness in small, semi-isolated groups for sports events” is well-organized and the significance of the main ideas are attractive. So, I suggest that this paper can be accepted to publish in "PLOS ONE".

7. PLOS authors have the option to publish the peer review history of their article (what does this mean?). If published, this will include your full peer review and any attached files.

Reviewer #1: No

Reviewer #2: No

---

## [Editor Report · Acceptance letter]

21 Mar 2022

PONE-D-21-32760R1 

COVID-19 testing systems and their effectiveness in small, semi-isolated groups for sports events 

Dear Dr. Kamo:

I'm pleased to inform you that your manuscript has been deemed suitable for publication in PLOS ONE. Congratulations! Your manuscript is now with our production department. 

Kind regards, 

on behalf of

Dr. Asep K. Supriatna 

Academic Editor

PLOS ONE